# The Importance of Measuring Mental Wellbeing in the Context of Psychological Distress: Using a Theoretical Framework to Test the Dual-Continua Model of Mental Health

**DOI:** 10.3390/bs13050436

**Published:** 2023-05-22

**Authors:** Jackson Mason Stephens, Matthew Iasiello, Kathina Ali, Joep van Agteren, Daniel B. Fassnacht

**Affiliations:** 1College of Education, Psychology and Social Work, Flinders University, Bedford Park, SA 5042, Australia; 2School of Psychological Science, University of Western Australia, Crawley, WA 6009, Australia; 3Mental Health and Wellbeing Theme, South Australian Health and Medical Research, Adelaide, SA 5000, Australia; 4School of Health, University of the Sunshine Coast, Sippy Downs, QLD 4556, Australia

**Keywords:** mental wellbeing, psychological distress, dual-continua model

## Abstract

The dual-continua model of mental health suggests that psychological distress and mental wellbeing operate on two distinct yet related continua, both uniquely contributing to overall mental health. Previous literature provides support for the dual-continua model; however, inconsistent methodologies lacking a common theoretical underpinning have led to findings that are difficult to compare across studies. Using archival data, this study aimed to test the following three theoretically derived criteria proposed to accurately examine the dual-continua model: (1) confirming independent existence, (2) disconfirming bipolarity, and (3) quantifying functional independence. Method: In total, 2065 participants (female *n* = 1193; 57.8%) completed two online assessments (minimum 30 days apart) measuring psychological distress, mental wellbeing, and demographic information. Results: In total, 1.1% of participants experienced high distress as well as mental wellbeing confirming that psychological distress and mental wellbeing exist independently (Criterion 1). Bipolarity (Criterion 2) was partly disconfirmed: mental wellbeing consistently decreased as symptom severity increased for depression; however, anxiety and stress did not meet bipolarity requirements. Functional independence (Criterion 3) was established: longitudinal analysis found that participants reliably and simultaneously increased (2.7%) or decreased (4.2%) in distress and mental wellbeing, while cross-sectional analysis showed that psychological distress only explained 38% of the variance in mental wellbeing. Discussion: Findings provide further support for the dual-continua model through analysis of the proposed assessment criteria, suggesting a further need to measure the dual-continua model at the subdomain level, e.g., depression, anxiety, and stress, as opposed to global psychological distress. Validation for the proposed assessment criteria provides important methodological foundations for future studies.

## 1. Introduction

Historically, the concept of mental health has been conceptualised as the absence of symptoms of mental illness or psychological distress without considering positive elements of mental health [1]. Here, mental health is assumed to operate on a unidimensional continuum, i.e., ranging from one extreme to the other, whereby the amelioration of symptoms of illness leads individuals towards a normative and healthy mental state of ‘feeling well’, commonly considered as ‘mental wellbeing’ [1,2,3]. Scholarly criticism of this unidimensional model was documented as early as 1958 by Marie Jahoda [4], who argued that a positive state of mental health could not be reduced to a lack of psychopathology. Consequently, decades of research into states of wellbeing slowly followed, resulting in the development of two key theoretical constructs. The first, *hedonic wellbeing*, refers to the attainment of pleasure, life happiness, and satisfaction [5,6]. The second, *eudemonic wellbeing*, refers to the realisation of life meaning, personal fulfilment, and self-actualisation [7,8]. Keyes [9] later added *social wellbeing*, one’s ability to prosper in their social context, as a third component of mental wellbeing, to be followed by a plethora of theories resulting from the advent of positive psychology from the early 2000s onwards [10].

Whilst an abundance of studies investigating mental illness and mental wellbeing have independently progressed our understanding of the human experience, it was not until more recently that scholars began to explore how “symptoms” of illness and wellbeing might interact to collectively contribute to one’s overall mental health [11]. Such enquiry ultimately led to criticism of the implicitly applied and largely untested bipolar conceptualisation, whereby mental illness seamlessly flows into mental wellbeing on a single continuum. Accordingly, several alternative explanations were offered, with the dual-continua model of mental health, as outlined by Keyes [12,13], being one of the most prominent alternative models.

### 1.1. Dual-Continua Model of Mental Health

The dual-continua model integrates the literature on clinical psychology and contemporary positive psychology, suggesting that symptoms of mental illness and mental wellbeing exist on two separate yet related continua (Figure 1). The model proposes that high levels of mental illness and mental wellbeing are coincidentally possible and that mental wellbeing can be improved despite the persistent presence of a mental illness [3,14]. Such a model demonstrates the need to measure both psychological distress and mental wellbeing together if one wishes to accurately understand and subsequently address an individual’s overall mental health status.

There is growing evidence in support of the dual-continua model as a more accurate account of overall mental health. A scoping review by Iasiello et al. [16] assessed a wide variety of clinical and non-clinical studies in both Western and non-Western populations, concluding that the dual-continua model holds greater explanatory power over the traditional bipolar approach. Here, evidence that supported the dual-continua model was found in 82 of the 83 studies, which is continuously strengthened by an ongoing body of research [17,18,19].

The studies identified by Iasiello et al. [16] utilised four different methods for testing the dual-continua model (i.e., confirmatory factor analysis, validation of subgroups within the dual-continua model, longitudinal observational designs, and the assessment of differential predictors and correlation with key outcome variables). While from a scientific point of view, it is desirable to find support in favour of a model or theory using distinct methodologies [20], the lack of a common theoretical underpinning is of particular concern and can create challenges in comparing findings between studies, subsequently fuelling more, rather than less, scepticism among academics about the model’s validity. For example, studies aiming to validate the existence of subgroups within the dual-continua model can use descriptors, cut offs, and terminology that varies depending on the theoretical background chosen by the authors [16]. Crucially, the abovementioned theoretical variation causes researchers to draw differing conclusions. For example, one study investigating a sample of Canadian students concluded evidence for *the dual-continua model* after 3.4% of students reported simultaneously high mental wellbeing and high psychological distress [21]. Conversely, a study by Zhao and Tay [22] found that 9% of participants reported “near-average” levels of mental wellbeing whilst suffering from moderate depression, citing a possible indication of coinciding mental wellbeing and psychological distress yet concluded evidence for *the traditional bipolar model*. These two studies highlight how simple it is to draw opposite conclusions about the dual-continua model’s validity based on interpretation by individual researchers. Although this problem is well reported within scientific research [23], it is, nonetheless, undesirable, and in the case of ‘bipolarity’ and dual-continua models can be reduced if researchers utilise a set of common criteria that guide scientific decision making.

### 1.2. What Assumptions Underpin Bipolarity in Mental Health?

We argue that it pays to interrogate the concept of bipolarity. It not only helps to formulate a stronger understanding of the opposing bivariate (dual-continua) perspectives but also allows us to better answer the question of what can be considered evidence for the dual-continua model. Cacioppo and Berntson [24] made a similar inquest regarding the nature of attitudes, which were originally assumed to be reducible to the net difference between one’s desire or aversion toward a stimulus, i.e., represented on a bipolar continuum. Cacicioppo and Berntson outlined three key assumptions essential to the bipolar perspective, which (if disconfirmed) could provide evidence that positive and negative attitudes are better represented via bivariate conceptualisations. The principles outlined by Cacioppo and Berntson are theoretically applicable to any investigation of bipolarity and thus have been used herein to form the basis for three theoretically supported criteria for testing the dual-continua model in any given population (see Iasiello et al. [25] for further discussion on this topic).

#### 1.2.1. Criterion 1: Confirming Independent Existence

First, Cacioppo and Berntson’s [24] *principle of evaluative activation* suggests that an attitude is the joint function of positive and negative activation, ranging from maximally positive (and minimally negative) to maximally negative (and minimally positive). In the mental health context, evaluative activation suggests that overall mental health is assumed to be reducible to the net difference between states of illness and mental wellbeing. For simplicity’s sake, we will use psychological distress as a worked example. For instance, “I am happy; however, I am more sad than happy; therefore, I am mildly sad”. Essential to the principle of evaluative activation for dual-continua models is to test that psychological distress and mental wellbeing are, in fact, two independently definable constructs. This forms the basis of Criterion 1, in which a study aims to establish that mental wellbeing is not merely the absence of symptoms of distress (or vice versa) from gaining evidence that groups thought to be impossible under a bipolar conceptualisation (i.e., those with high psychological distress and high mental wellbeing) do, in fact, exist.

#### 1.2.2. Criterion 2: Disconfirming Bipolarity

Second, Cacioppo and Berntson’s [24] *principle of opposing evaluative activations,* extending the principle of evaluative activation, suggests that positive and negative activation have opposing effects on an attitude. Here, ‘units’ of psychological distress and mental wellbeing are equal, each cancelling the other out to provide an overall equation of one’s overall mental health status. Criterion 1, evidence of independent existence, does not contradict the possibility that psychological distress and mental wellbeing operate on a single dimension separated *into two parts*. In this case, an increase in one construct can equate to an equal and opposite decrease in the other, meaning that the dual-continua conceptualisation would hold no additional explanatory power over bipolar perspectives. Accordingly, Criterion 2 aims to examine whether psychological distress and mental wellbeing exist at the extreme ends of a *single continuum* by assessing the pattern of mental wellbeing at various severity levels of psychological distress.

#### 1.2.3. Criterion 3: Quantifying Functional Independence

The *principle of reciprocal evaluative activation* suggests that positive and negative activations are reciprocally controlled. That is, an increase in psychological distress automatically evokes an *equal and opposite* decrease in mental wellbeing, which can feasibly be assessed by interrogating differential changes in psychological distress and mental wellbeing over time, as well as the variance in mental wellbeing explained by psychological distress. On the relationship between positive and negative aspects of overall mental health, Cacioppo and Berntson [24] endorse a *functional independence* approach as opposed to a *stochastic independence* one. *Stochastic independence* describes a relationship between two constructs whereby the likelihood of their joint occurrence is equal to the likelihood of their independent occurrences [24]. In the dual-continua model context, stochastic independence suggests that psychological distress and mental wellbeing are completely unrelated constructs, which is evidently not the case [16], nor does it need to be in order for the dual-continua model to establish itself as a more informative model. Alternatively, *functional independence*, i.e., instances where one construct does and the other does not fluctuate as the result of a third independent variable, presents as a more relevant independence principle in the case of mental health. Accordingly, Criterion 3 is not to establish complete separability (stochastic independence) but rather to identify circumstances where psychological distress is, and is not, paired with an equal and opposite manifestation of mental wellbeing (and vice versa). That is to say, if not completely separate, how separate are the two continua?

### 1.3. A Framework to Test Dual-Continua Models

The above outlined three criteria—(1) *confirming independent existence*, (2) *disconfirming bipolarity*, and (3) *quantifying functional independence*—can have utility as a guiding framework to establish higher quality evidence for or against dual-continua models, as the evidence quality to date has scope for improvement. Such clear conceptualisation may not have been necessary when the dual-continua model was first introduced by Keyes [12,13], as only basic understanding was required to open a new area of the inquest. However, a growing body of evidence supporting the model calls for added clarity and precision when investigating the model. Establishing a set of criteria for measurement of the dual-continua model is a necessary first step to avoid scientific miscommunication.

Published literature to date only provides us with a limited number of examples that have *implicitly* applied the three abovementioned criteria. For example, in a study examining the effectiveness of Acceptance and Commitment Therapy (ACT) on psychological distress and mental wellbeing outcomes, Trompetter et al. [26] found that two-thirds to three-quarters of participants who improved during ACT improved on either psychological distress or mental wellbeing measures. Such results provide evidence disconfirming bipolarity (i.e., Criterion 2) and also implicitly quantify functional independence (i.e., Criterion 3) by showing instances where one construct does and another does not vary as the result of a third independent variable (ACT). Similarly, de Vos et al. [27] found evidence for independent existence (i.e., Criterion 1) and disconfirming bipolarity (i.e., Criterion 2), with 13% of participants displaying high levels of mental wellbeing despite the diagnosis of a mental illness (e.g., an eating disorder). Furthermore, de Vos et al. [27] also found evidence to quantify functional independence (i.e., Criterion 3), finding that personality-related factors such as emotional regulation significantly predict mental wellbeing outcomes but do not predict eating disorder psychopathology.

These limited examples apart, existing research tends to fall short of testing each of the three criteria. This can partly be explained by the choice of methodology, with many studies utilising a cross-sectional design [16], which per definition, is not able to find evidence for criterion 3, as it requires follow-up measures to be conducted.

### 1.4. Aims of this Current Study

The aim of this current study is to explicitly test the dual-continua model using three theoretically derived criteria in a large community sample. By doing so, it aims to illustrate the utility of the three criteria (as derived from Cacioppo and Berntson’s [24] work) and to provide an empirical example of how the framework can be used to find evidence for dual-factor models of mental health.

It is predicted that a significant proportion of participants will report simultaneously high levels of psychological distress and mental wellbeing. Additionally, it is predicted that mental wellbeing will not significantly decrease at each categorical increase in distress severity. Furthermore, it is predicted that a significant proportion of participants will increase in wellbeing and increase in distress and conversely decrease in wellbeing and decrease in distress from their first to second assessment. Finally, it is predicted that levels of psychological distress will not predict variance in mental wellbeing above what could be expected under the dual-continua model conceptualisation.

## 2. Materials and Methods

This current study used data collected by the South Australian Health and Medical Research Institute (SAHMRI; Adelaide, South Australia) between March 2019 and July 2022, a subset of which has been used in a previous study, see Iasiello et al. [28].

### 2.1. Recruitment and Participants

Participants were adults from the general population who utilised services provided by SAHMRI, including internet-based mental health and wellbeing measurements and non-specific mental health interventions. Recruitment in this present study occurred through three avenues. First, individuals could self-enrol via a website that focuses on mental health and wellbeing, which the authors have been running for a number of years. Second, participants could be part of wellbeing measurements that SAHMRI conducts with community organisations and workforces. Third, participants could complete measures as part of a baseline assessment for an online mental health and wellbeing program (*Be Well Plan*). In all instances, participants who completed as many measurements as they wanted online in their own time were automatically provided with a tailored mental health report explaining their individual scores and were not compensated for participation. Additional information was also provided, including resources to improve mental health and information about available mental health services if needed.

From an initial dataset of 18,680, a sub-sample of 2815 participants who completed at least two measures of both psychological distress and mental wellbeing was drawn. Participants who (1) were under the age of 18 or did not provide age information (*n* = 38), (2) completed measures whilst outside Australia (*n* = 30), or (3) did not complete the necessary questions to calculate total wellbeing, and/or distress score for at least two separate measurements (*n* = 45) were excluded. To allow adequate time for participants to show reliable changes in mental health from their first to second assessment, participants who completed measures less than 30 days apart (*n* = 637) were also excluded.

The final Australian sample (*N* = 2065) were predominantly female individuals (*n* = 1193; 57.8%) with an average age of 37.9 years (*SD* = 13.4). The majority of participants were employed (*n* = 1431; 69.3%), 310 (15%) were studying, and 37 (1.7%) were semi or fully retired when they completed their first measure. The number of days between participants’ first and second assessments ranged from 30 to 1146 days (*M* = 168.4, *SD* = 170.1, *Mdn* = 120.0).

### 2.2. Materials

Demographics. Participants were asked about their age, gender, location during survey completion, current employment status, current study status, as well as whether they were currently either semi or fully retired.

Psychological Distress. Psychological distress was measured using the 21-item Depression Anxiety and Stress Scale (DASS-21) [29]. The DASS-21 uses a 4-point Likert scale ranging from 0 (*never*) to 3 (*almost always*) to measure psychological distress, namely depression, anxiety, and stress, over the past week, with higher scores indicating greater symptom severity. The DASS-21 provides reliable symptom severity cut-off scores for depression (normal = 0–9, mild = 10–13, moderate = 14–20, severe = 21–27, extremely severe = 28+), anxiety (normal = 0–7, mild = 8–9, moderate = 10–14, severe = 15–19, extremely severe = 20+), and stress (normal = 0–14, mild = 15–18, moderate = 19–25, severe = 26–33, extremely severe = 34+) [29]. Internal consistencies of the DASS-21 ranged from good to excellent at both the first (total α = 0.94; depression α = 0.92; anxiety α = 0.85; stress α = 0.87) and second assessment (total α = 0.95; depression α = 0.92; anxiety α = 0.87; stress α = 0.88) in this present study.

Mental Wellbeing. Mental wellbeing was measured using the Mental Health Continuum Short-Form (MHC-SF) [30]. Participants respond to 14 items relating to their feelings over the past month on a 6-point Likert scale ranging from 0 (*never*) to 5 (*everyday*), with higher scores indicating greater levels of wellbeing. The MHC-SF provides an overall wellbeing score, allowing for categorisation into high (*flourishing*), moderate, or low (*languishing*) wellbeing. Furthermore, individual items measure the three domains of wellbeing outlined by Keyes et al. [30], providing subscale scores of emotional (e.g., “how often did you feel happy?”), psychological (e.g., “how often did you feel that your life has a sense of direction and meaning?”), and social (e.g., “how often did you feel that people are basically good?”) wellbeing. The internal consistencies of the measure ranged from acceptable to good at the first (total α = 0.94; emotional α = 0.87; psychological α = 0.88; social α = 0.85) and second assessment (total α = 0.95; emotional α = 0.90; psychological α = 0.90; social α = 0.88) in this present study.

### 2.3. Statistical Analysis

*Classification of Distress.* This present study utilised two classifications of distress: high versus low distress for chi-square analysis and symptom severity groups for Kruskal–Wallis H tests: For Fisher’s exact test, participants were categorised as *high distress* if they met the DASS-21 cut-off scores indicating *severe* or *extremely severe* symptom severity for depression, anxiety, and/or stress. Participants were categorised as *low distress* if symptom severity was *normal*, *mild*, or *moderate* on each of the depression, anxiety, and stress subscales. Further analysis was conducted whereby participants with *moderate* symptom severity on the depression, anxiety, or stress subscale were categorised in the *high-distress* group. For Kruskal–Wallis H tests, participants were classified as having *normal*, *mild*, *moderate*, or *severe* symptom severity for depression, anxiety, and stress according to the DASS-21 subscale cut-off scores. Due to the relatively small number of participants experiencing *extremely severe* depression (6.6%), anxiety (10.2%), and stress (1.9%), the *severe* and *extremely severe* categories were combined to create one category, indicating *severe* symptom severity for each DASS-21 subscale.

*Classification of Wellbeing.* Classification of *flourishing*, moderate, or *languishing* mental wellbeing was calculated using the MHC-SF. Here, participants were categorised as *flourishing* if they had responded with a response of *every day* or *almost every day* to at least one (out of three) item measuring hedonic wellbeing (items 1–3) and at least six (out of 11) items measuring positive functioning (items 4–14) [30]. Participants were categorised as *languishing* if they responded with a response of *never* or *once or twice*, to at least one item measuring hedonic wellbeing, and at least six items measuring positive functioning. If participants did not meet criteria for either *flourishing* or *languishing*, they were categorised into the *moderate* wellbeing category.

*Testing Criterion 1.* Fisher’s exact test using participants’ first measure was conducted to test whether the observed frequency of participants reporting psychological distress whilst *flourishing* was significantly different from the expected frequency under the assumption of one bipolar continuum (*n* = 0, *flourishing* with high psychological distress).

*Testing Criterion 2.* As the DASS-21 does not provide severity cut-off scores for total distress, symptom severities for the three subscales were analysed. Data distribution did not meet normality assumptions; therefore, three separate Kruskal–Wallis H tests were used to compare median wellbeing scores as symptom severity increased on the three subscales of distress, i.e., median wellbeing at normal, mild, moderate, severe, and extremely severe symptoms severity for each subscale.

*Testing Criterion 3.* The functional independence of psychological distress and mental wellbeing was tested both longitudinally and cross-sectionally. First, Reliable Change Indices (RCI) were calculated to determine whether participants demonstrated a meaningful change in distress and/or wellbeing from their first to second measure. Specifically, RCI scores +/− 1.96 standard deviations indicated that the participant’s change in outcome was statistically significant [31]. Second, the variance in mental wellbeing explained by psychological distress was tested using a hierarchical regression. Demographic information was added before the total DASS-21 score.

Significance was determined at an alpha level of 0.05 for *p* values. Interpretation of Cramer’s V effect size was based on Cohen’s [32] benchmarks. Cramer’s V with 1 degree of freedom was interpreted as follows: V = 0.10 for small, V = 0.30 for moderate, and V = 0.50 for large. Cramer’s V with 3 degrees of freedom was interpreted as follows: V = 0.06 for small, V = 0.17 for moderate, and V = 0.29 for large. A Kruskal–Wallis H test was used to compare group median scores, as data did not meet assumptions of normality, while all other assumptions were met. Due to the large sample size used, statistical power for analysis was deemed sufficient.

## 3. Results

Prior to testing the utility of three theoretically driven criteria designed to clearly outline the evidence for or against the dual-continua model, descriptive statistics were calculated, and participants were classified into high and low levels of distress, as well as *languishing*, moderate, and *flourishing* levels of mental wellbeing, see Table 1.

### 3.1. Criterion 1: Independent Existence

As seen in Table 2, 23 participants (1.1%) were classified as *flourishing* with high psychological distress. Fisher’s exact test, χ^2^(1, *N* = 954) = 23.57, *p* < 0.001, Cramer’s V = 0.16, suggested that these participants represent a significant proportion of the sample (small to medium effect). Furthermore, when *moderate* symptom severity was included in the *distress* categorisation, the number of participants classified as *flourishing* with psychological distress increased to 88 (4.3%), representing a significant proportion of the sample, χ^2^(1, *N* = 911) = 92.47, *p* < 0.001, Cramer’s V = 0.23 (small to medium effect).

### 3.2. Criterion 2: Disconfirming Bipolarity

This study yielded different results depending on the type of distress analysed. Comparing levels of mental wellbeing for each symptom, the severity level of depression revealed statistically significant differences, *H*(3) = 936.00, *p* < 0.001. Follow-up post hoc analysis showed that the differences occurred in the pattern predicted by a bipolar model; median mental wellbeing significantly decreased at each depression severity increase from normal to severe as follows (see Figure 2a): normal (*Mdn* = 54.0), mild (*Mdn* = 44.0), moderate (*Mdn* = 39.0), and severe (*Mdn* = 26.0).

Comparing levels of mental wellbeing for each symptom, the severity level of anxiety once again revealed statistically significant differences, *H*(3) = 469.11, *p* < 0.001. Follow-up post hoc analysis showed that significant differences in wellbeing occurred between normal (*Mdn* = 53.0) to mild (*Mdn* = 45.0) and from moderate (*Mdn* = 43.0) to severe (*Mdn* = 34.0) anxiety severity. However, there was no significant difference in mental wellbeing between mild to moderate anxiety (*p* = 0.915; see Figure 2b).

Similarly for stress, comparing levels of mental wellbeing for each symptom severity level revealed statistically significant differences, *H*(3) = 481.30, *p* < 0.001. Follow-up post hoc analysis showed that the differences in mental wellbeing were significant between normal (*Mdn* = 52.0) to mild (*Mdn* = 43.0) and moderate (*Mdn* = 40.0) to severe (*Mdn* = 32.0) stress severity. However, there was no significant difference in mental wellbeing between mild to moderate stress severity (*p* = 0.093; see Figure 2c).

### 3.3. Exploratory Analysis

As findings suggested that depression followed a bipolar relationship with mental wellbeing while anxiety and stress did not, we conducted additional analyses. Hierarchical linear regression was performed to analyse the variance in total mental wellbeing explained by the three subcategories of distress (i.e., depression, anxiety, and stress). After controlling for age, gender, and work/student status, participants’ scores for depression, anxiety, and stress were added. The overall regression was significant, *R*^2^ = 0.561, *F*(8, 2056) = 328.47, *p* < 0.001, adjusted *R*^2^ = 0.559. The addition of depression, anxiety, and stress into the model explained a significant 47.1% of the additional variance. Critically, the vast majority of this additional variance was explained by the depression (β = −0.688) subscale, with anxiety (β = 0.041) and stress (β = −0.080) not providing any noteworthy additional predictive power over mental wellbeing.

### 3.4. Criterion 3: Quantifying Functional Independence

Longitudinal Analysis using Reliable Change Index (RCI). As seen in Table 3, 56 participants (2.7%) simultaneously increased in distress and mental wellbeing; conversely, 87 participants (4.2%) decreased in distress and mental wellbeing; these combinations would not exist according to the bipolar model. Fisher’s exact test indicated that the proportion of participants identified in either of these categories was above what could be expected by chance, χ^2^ (3, *N* = 1524) = 159.00, *p* < 0.001. Cramer’s V = 0.32, indicating a large effect.

Cross-sectional Analysis using Hierarchical Linear Regression. After controlling for age, gender, and work/student status in Step 1, the total DASS-21 score explained an additional 38% of the variance in mental wellbeing, *F*_change_ (1, 2058) = 1447.76, *p* < 0.001, leaving 62% of unexplained variance.

## 4. Discussion

This present study aimed to demonstrate the utility of a set of three theoretically derived criteria aimed at examining evidence for or against bipolarity—the common assumption that psychological distress and mental wellbeing fall at polar opposites of a single continuum—or for or against the opposing dual-continua models of mental health. The results showed that (1) psychological distress and mental wellbeing existed as separate constructs, (2) depression and mental wellbeing operated on a bipolar continuum, while anxiety and stress did not, and importantly, (3) psychological distress and mental wellbeing functioned independently between time points. The results are first discussed, followed by the implications of the findings and the utility of the three tested criteria within practice and mental health research.

While the results themselves are not necessarily novel and have been observed in previous research, these studies often relied on only one of these criteria to disconfirm bipolarity and thus support the dual-continua model of mental health.

### 4.1. Independent Existence

As predicted, Criterion 1 was supported by both mental wellbeing and distress, simultaneously manifesting at high levels. A small yet significant proportion (1.1%) of participants were seen to be *flourishing with high psychological distress*. These findings are consistent with previous literature showing a similar proportion (around 1.5%) of individuals with simultaneously high symptoms of illness and mental wellbeing [14]. Furthermore, the number of individuals *flourishing with psychological distress* increased to 4.3% when participants with *moderate* symptom severity were included in the *distress* categorisation. Here, the results highlight the impact of arbitrary categorisation on subsequent findings in relation to the dual-continua model, suggesting a need to consider practical implications when determining categorisation cut-offs in any given study [33]. Regardless, both findings would not be possible if the presence of one construct equates to the absence of the other, as suggested by traditional unipolar perspectives. Theoretically, no study may conclude the evidence for the dual-continua or bipolar models without (at least implicitly) satisfying the criteria of independent existence. For example, studies using confirmatory factor analysis demonstrate independent existence via findings that the dual-continua model displays a better statistical fit than bipolar models [34,35]. While satisfaction for Criterion 1 does not provide evidence for *or* against bipolarity, establishing independent existence is an important prerequisite to dispute traditional unipolar models, where only psychological distress exists, and mental wellbeing is simply the absence of distress.

### 4.2. Disconfirming Bipolarity

Criterion 2 was partly supported. An increase in symptom severity from mild to moderate levels of anxiety and stress was not reciprocated by a statistically significant decrease in mental wellbeing, as a bipolar model suggests. However, the depression subscale did follow a bipolar pattern in that mental wellbeing significantly decreased with every increase in the severity of depressive symptoms.

The lack of evidence to disconfirm bipolarity between depression and mental wellbeing is consistent with previous research; for example, strong negative correlations between subjective wellbeing and depression or mood disorders [36,37,38]. Here, the observed bipolar relationship may be due to intrinsic similarities between conceptualisations of psychological distress, i.e., depressive symptoms, and mental wellbeing. Given that psychological research was firmly rooted in psychopathology prior to the emergence of positive psychology, it is likely that mental wellbeing research unintentionally—and in some cases intentionally, e.g., [39]—generated clusters of wellbeing symptoms mirroring those already established in the *Diagnostic and Statistical Manual of Mental Disorders* (DSM-5) [12]. For example, *depression* involves the presence of negative affect, anhedonia, and a lack of positive affect, while *emotional wellbeing* involves positive affect, hedonia, and the absence of negative affect [12,39]. As a result of such opposite conceptualisations, fundamentally antonymous measurement items are used to assess such constructs. For example, items such as “*I couldn’t seem to experience any positive feeling at all*” are used to measure depression, while items such as “*How often did you feel happy*” are used to measure emotional wellbeing. Accordingly, it is unsurprising that high levels of depression are highly correlated with low mental wellbeing and vice versa.

The finding that bipolarity was disconfirmed for anxiety is consistent with previous research. For example, Trompetter et al. [26] found that 72% of participants who completed Acceptance and Commitment Therapy (ACT) improved on *either* anxiety symptoms *or* positive mental health outcomes, suggesting that anxiety and mental wellbeing operate on two functionally independent continua. Notably, mixed evidence has been found for the dual-continua model in severe clinical populations [16]. For example, van Erp Taalman Kip and Hutschemaekers [40] found strong negative correlations between psychological distress (including anxiety) and mental wellbeing in a clinical sample. Similarly, we found a significant decrease in mental wellbeing when psychological distress increased from moderate to severe severity across all subdomains of psychological distress. Here, the presence of severe distress may prevent mental wellbeing from manifesting; thus, the two constructs are ‘more bipolar’ when severe distress is experienced.

The finding that stress and mental wellbeing do not operate on a bipolar continuum is consistent with previous studies. For example, Díaz et al. [41] found that post-traumatic stress and mental wellbeing were best represented on two closely related yet distinct continua. Furthermore, Jovanovic and Brdaric [42] found that mental wellbeing and stress were differentially predicted by high curiosity; in their study, stress was only moderately correlated with aspects of mental wellbeing, including life satisfaction, hope, and purpose in life. Hence, our results relating to stress concur with the available studies, further substantiated by our exploratory finding that the stress subscale explained only a negligible amount of variance in mental wellbeing.

### 4.3. Functional Independence

As predicted and consistent with previous research, e.g., [26,43], Criterion 3 was supported by psychological distress and mental wellbeing seen to operate (partly) independently. That is, changes in distress and mental wellbeing were seen to occur outside of the equal and opposite pattern expected under the bipolar conceptualisation. Similarly, psychological distress explained only a modest amount of variance in mental wellbeing providing further evidence for functional independence.

For depression—despite explaining more variance in wellbeing than anxiety and stress—the relationship with mental wellbeing was still seen to be functionally independent in this present study. This finding theoretically contradicts our earlier result suggesting that depression and mental wellbeing operate on a bipolar continuum, as bipolar assumptions negate the possibility of functional independence [24,38]. Importantly, this finding suggests that cross-sectional observations of bipolarity do not necessarily disqualify functional independence. Rather, it is possible that while psychological distress and mental wellbeing appear to operate in a bipolar pattern when intervened with and measured over time, their functional independence becomes apparent. If this is the case, then bipolar measurement likely provides an incomplete indication of overall mental health. Accordingly, future research should further investigate bipolarity in relation to the criteria of functional independence.

### 4.4. Implications and Future Directions

Our findings concur with previous literature [26,44,45], suggesting the need to measure both psychological distress and mental wellbeing to accurately determine one’s overall mental health. The inability to identify perfect bipolarity means data obtained through a bipolar conceptualisation, which solely assesses psychological distress, is inferior to that obtained via the utilisation of a dual-continua conceptualisation [46]. Whilst distress and mental wellbeing can appear to operate on a single bipolar continuum [22] and are often highly negatively correlated [14], their functional independence indicates that examining levels of one construct is not always sufficient for predicting levels on the other [25]. Furthermore, it is important to note that defining overall mental health via the measurement of psychological distress alone may result in a distorted perception of mental health at the individual and population level [44], similar to how the measurement of wellbeing using uni-dimensional scales such as life satisfaction would lead to distorted perceptions of wellbeing. Accordingly, evidence disconfirming perfect bipolarity should be considered an indication that measurement using the dual-continua conceptualisation will better serve academic progress within that given study field.

The findings that, overall, psychological distress and mental wellbeing were seen to operate on functionally independent dual continua support the notion that mental wellbeing can be independently enhanced, even in individuals experiencing a mental illness [27,47,48]. Thus, the degree of independence provides further support for claims that dual-continua model perspectives should be considered to underpin the healthcare system [16] and could be used to justify an uptake of salutogenic approaches within therapy and the broader healthcare system [49]. Whilst the adoption of a salutogenic approach may appear less urgent than the currently dominant method focussing on pathogenic factors, greater mental wellbeing can be protective against a future diagnosis of mental illness [50,51,52], may provide suicide resilience [53] and assist in mental illness recovery [54,55]. Additionally, it can be argued that ‘*flourishing*’ individuals are more productive in society, miss fewer days at work, have fewer health conditions leading to daily function impairment, and display higher psychosocial functioning [12]. Accordingly, given that less than half of the population is currently ‘*flourishing*’, e.g., [56], our results suggest the promotion of mental wellbeing initiatives represents a potentially productive yet currently unexhausted strategy to optimise mental health [49].

### 4.5. Study Strengths and Limitations

The use of a widely applicable, theoretically driven set of criteria to assess the dual-continua model is a strength of this present study that lays important methodological foundations for future research. With a growing body of evidence for the model, the ability to collate, compare, and contrast the literature is a vital step to optimising the potential benefits of such a model. Additionally, a large sample with a relatively even gender distribution is a strength of this present study as previous research often consisted of samples dominated by females [26,57,58]. Moreover, the use of a general community sample is both a strength, allowing for broadly applicable evidence for the model, and a limitation, as findings cannot be generalised to clinical samples, a population in which the dual-continua model may manifest less distinctly [40].

It is worth noting that the categorisation of high versus low distress in this present study differs from the various previously used methods [43,44]. However, cut offs used in this present study represent a more stringent approach; therefore, it is possible that we underestimated, as opposed to overestimated, the proportion of individuals who were *flourishing with a mental illness*.

## 5. Conclusions

This study highlights the utility of a widely applicable theoretical framework for measuring the dual-continua model of overall mental health, advancing a theoretical and methodological foundation for future research. By way of a worked example, the findings in this study suggest further evidence for the dual-continua model with practical implications on an individual and population level. Specifically, the results firstly suggest the need to measure both psychological distress and mental wellbeing to understand an individual’s overall mental health and support claims to integrate salutogenic approaches into the currently pathogenic-focused mental healthcare system. Secondly, the analysis of the three theoretically driven criteria suggests that dual-continua model research may be most fruitful at the subdomain level, where it is possible that multiple differential relationships exist between subdomains of mental wellbeing and psychological distress.

## Figures and Tables

**Figure 1 behavsci-13-00436-f001:**
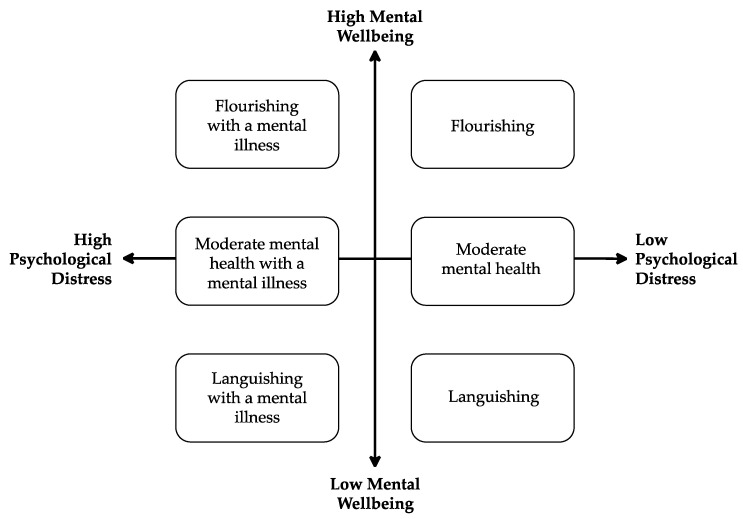
Visual Depiction of the Dual-Continua Model of Mental Health [15].

**Figure 2 behavsci-13-00436-f002:**
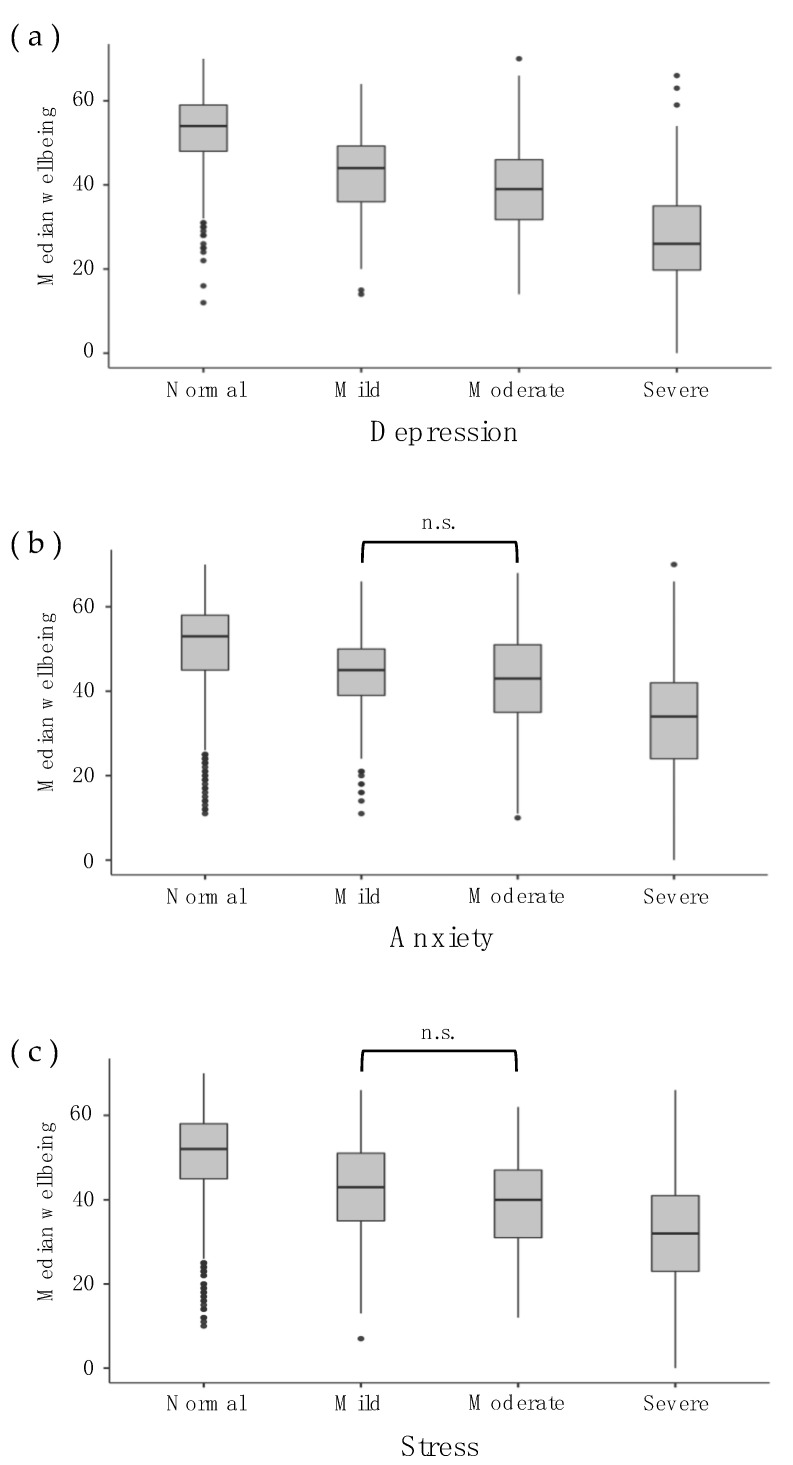
Median Mental Wellbeing (Total Score) at Each Symptom Severity Level of Distress in Each Distress Subdomain. Bracket indicating “n.s” refers to a non-significant decrease in mental wellbeing despite a categorical increase in psychological distress.

**Table 1 behavsci-13-00436-t001:** Means and Standard Deviations for Psychological Distress and Mental Wellbeing.

	Range	M	SD
DASS-21			
Total	0–126	30.24	23.73
Depression	0–42	9.39	9.52
Anxiety	0–42	7.60	8.07
Stress	0–42	13.26	9.03
MHC-SF			
Total	0–70	46.04	13.00

*Note*. DASS-21 = Depression Anxiety and Stress Scale, MHC-SF = Menatl Health Continuum–Short Form.

**Table 2 behavsci-13-00436-t002:** Number and Percentage of Participants Classified as Low vs. High Psychological Distress by Category of Mental Wellbeing.

	Psychological Distress		
Mental Wellbeing	Low	High	Total
Flourishing	661 (32.0%)	23 (1.1%)	684 (33.1%)
Moderate	924 (44.7%)	427 (20.7%)	1351 (65.4%)
Languishing	3 (0.1%)	27 (1.3%)	30 (1.5%)
Total	1588 (76.9%)	477 (23.1%)	2065 (100%)

**Table 3 behavsci-13-00436-t003:** Number of Participants Who Changed Reliably from Assessment 1 to Assessment 2 (*N* = 2065).

	Psychological Distress		
Mental Wellbeing	Increased	Decreased	No Change
Increased	56 (2.7%) ^1^	340 (16.5%)	351 (17.0%)
Decreased	279 (13.5%)	87 (4.2%) ^1^	266 (12.9%)
No Change	128 (6.2%)	145 (7.0%)	413 (20.0%)

^1^ Key values of interest for dual-continua model measurement.

## Data Availability

Data is available upon reasonable request.

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
