# Peer review of "The Importance of Measuring Mental Wellbeing in the Context of Psychological Distress: Using a Theoretical Framework to Test the Dual-Continua Model of Mental Health"

_behavsci, 2023, doi:10.3390/bs13050436_

Round 1
Reviewer 1 Report
Dear Editors and authors,
I thank you for my consideration as a reviewer of this manuscript. It is my pleasure to contribute to International Journal of Environmental Research and Public Health.
This research contributes to the knowledge of the relevance of mental well-being in psychological distress. The manuscript is well-justified, the size sample is adequate, and the results resolve the research questions.
I would like to suggest the followings comments:
I recommend writing the hypotheses or research questions of this study.
Regarding recruitment, I recommend including the dissemination of the study and whether participants were compensated for participation.
Concerning statistical analysis, I would like to ask the authors the reason for using a non-parametric test (e.g., Krukal-Wallis) and a parametric test (e.g., ANOVA).
Moreover, I recommend indicating the interpretation of the results in the discussion section; for this reason, I suggest eliminating them in the results section.
Author Response
Reviewer #1
This research contributes to the knowledge of the relevance of mental well-being in psychological distress. The manuscript is well-justified, the size sample is adequate, and the results resolve the research questions.
I would like to suggest the followings comments:
I recommend writing the hypotheses or research questions of this study.
RESPONSE 1: Thank you for this suggestion. We have added the section “1.4 Aims of the Current Study” to include the aim of the study and a hypothesis for each criteria.
Regarding recruitment, I recommend including the dissemination of the study and whether participants were compensated for participation.
RESPONSE 2: In accordance with the above suggestion, we have included additional information to section 2.1 Recruitment and Participants about the services provided by SAHMRI to address the dissemination of the study. Further, also in section 2.1, we have clarified that participants were not compensated for this participation in the study.
Concerning statistical analysis, I would like to ask the authors the reason for using a non-parametric test (e.g., Krukal-Wallis) and a parametric test (e.g., ANOVA).
RESPONSE 3: Thank you for requesting this information, we apologise for the lack of clarity. ANOVA assumptions were not met due to outliers, therefore only Kruskal-Wallis test was used. To clarify this, we have removed any mention of ANOVA tests, and instead stated that the use of a Kruskal-Wallis test was due to data not meeting assumptions of normality – pg.7, line 305 and pg.7, line 322.
Moreover, I recommend indicating the interpretation of the results in the discussion section; for this reason, I suggest eliminating them in the results section.
RESPONSE 4: Duly noted, thank you for bringing this to our attention. We have removed interpretation of results from results section.
Reviewer 2 Report
The topic is interesting. I just have one comment about the abstract. The authors should introduce the three criteria first and then demonstrate the results. That would be easier to follow for readers.
Author Response
Reviewer #2
The topic is interesting. I just have one comment about the abstract. The authors should introduce the three criteria first and then demonstrate the results. That would be easier to follow for readers.
RESPONSE 4: Thank you for your comment. We have changed the wording in abstract to explicitly introduce “the three criteria” before stating the results for each criteria.
Reviewer 3 Report
Thank you for the opportunity to revise this manuscript.
As a general impression, I believe this study carries an underlying theoretical relevance with respect to the need to accurately disentangle the interactions between mental well-being and mental distress. I commend the authors for their attempt to contribute to this inquiry.
Nevertheless, I have several concerns about the way this question was addressed in this study, which make me think about the potential contribution of the present findings. To make this review helpful, I will focus on the two main (highly related) limitations that I see.
To inspect whether mental health is better conceptualized following a bipolar perspective, the authors applied three criteria provided by Caccioppo and Berntson. As it is presented, the introduction of the three criteria seems vaguely connected to the formulation of this question. Please do not misunderstand this comment - I do not pretend to say that the arguments are inappropriate. Rather, for a general readership who is unfamiliar with Cacioppo and Berntson's paper, it might seem difficult to understand why these criteria are used to explore the dual-continua model of mental health. As the authors stated that it is "a widely applicable" set of criteria, it is important to accommodate and probe this idea (which, in its current form, is lacking in the manuscript). Therefore, I really recommend the authors dig deeper into the rationale that makes these criteria a useful framework to disentangle the structure of mental health following the dual-continua model.
Connected with the previous comment, I struggle to see the contribution that the empirical data offers (compared to the existing evidence). The distribution of "mentally ill but flourishing" individuals is not a novel finding (it was even presented in the former papers of the MHC). I support the replication of such findings, but within the scope of the present study, it seemed like the authors "transferred" a theoretical framework in their introduction to test the bipolarity concept of mental health, but the analytical strategy (data collection and analysis) did not differ from previous studies that did not use such theoretical framework. This yields the impression that the theoretical framework seems not a fundamental requisite to explore this question. To sum up, it is important to offer a clearer conceptualization of how (and why) this theoretical approach can entail an advancement in the empirical research of the dual-continua model of mental health. I encourage the authors not to abandon their efforts in this attempt.
Author Response
Reviewer #3
Thank you for the opportunity to revise this manuscript.
As a general impression, I believe this study carries an underlying theoretical relevance with respect to the need to accurately disentangle the interactions between mental well-being and mental distress. I commend the authors for their attempt to contribute to this inquiry.
Nevertheless, I have several concerns about the way this question was addressed in this study, which make me think about the potential contribution of the present findings. To make this review helpful, I will focus on the two main (highly related) limitations that I see.
To inspect whether mental health is better conceptualized following a bipolar perspective, the authors applied three criteria provided by Caccioppo and Berntson. As it is presented, the introduction of the three criteria seems vaguely connected to the formulation of this question. Please do not misunderstand this comment - I do not pretend to say that the arguments are inappropriate. Rather, for a general readership who is unfamiliar with Cacioppo and Berntson's paper, it might seem difficult to understand why these criteria are used to explore the dual-continua model of mental health. As the authors stated that it is "a widely applicable" set of criteria, it is important to accommodate and probe this idea (which, in its current form, is lacking in the manuscript). Therefore, I really recommend the authors dig deeper into the rationale that makes these criteria a useful framework to disentangle the structure of mental health following the dual-continua model.
RESPONSE 5: Thank you for your careful review of our manuscript and productive feedback. To outline our rationale for using Caccioppo and Bertson’s work as a guiding principle for our three criteria, we have included more contextual information about Caccioppo and Berntson’s investigation of attitudes (within section 2.1 What Assumptions Underpin Bipolarity in Mental Health). We hope this additional information provides a clearer connection between our investigation of bipolarity within a mental health context, and Caccioppo and Berntson’s parallel investigation of positive and negative attitudes.
Connected with the previous comment, I struggle to see the contribution that the empirical data offers (compared to the existing evidence). The distribution of "mentally ill but flourishing" individuals is not a novel finding (it was even presented in the former papers of the MHC). I support the replication of such findings, but within the scope of the present study, it seemed like the authors "transferred" a theoretical framework in their introduction to test the bipolarity concept of mental health, but the analytical strategy (data collection and analysis) did not differ from previous studies that did not use such theoretical framework. This yields the impression that the theoretical framework seems not a fundamental requisite to explore this question. To sum up, it is important to offer a clearer conceptualization of how (and why) this theoretical approach can entail an advancement in the empirical research of the dual-continua model of mental health. I encourage the authors not to abandon their efforts in this attempt.
RESPONSE 6: Once again, thank you for bringing this to our attention. To address the comment above we have altered section 1.1 (line 82) to focus more clearly on theoretical underpinnings rather than methodology so as not to distract the reader from the key study aims. We hope that this change helps to emphasise that the criteria are not designed to decipher what statistical analyses are appropriate, but rather act as a theoretical framework to guide how evidence for the model should be sought and confirmed.
Further, we have added to section 3.1 to more clearly explain why investigation of the dual-continua model via the three criteria is necessary. We also hope that the addition of clear study aims, and the additional information provided on Cacioppo and Berntson’s work (as mentioned in response 5) will contribute to the reviewer’s concerns relating to the contribution of our three criteria approach.